# Effectiveness of a nurse-led hospital-to-home transitional care intervention for older adults with multimorbidity and depressive symptoms: A pragmatic randomized controlled trial

**Maureen Markle-Reid**[1ᴑ]*, **Carrie McAiney**[2ᴑ], **Kathryn Fisher**[1ᴑ], **Rebecca Ganann**[1ᴑ], **Alain P. Gauthier**[3‡], **Gail Heald-Taylor**[1‡], **Janet E. McElhaney**[4‡], **Fran McMillan**[5‡], **Penelope Petrie**[1‡], **Jenny Ploeg**[1‡], **Diana J. Urajnik**[5‡], **Carly Whitmore**[1‡]

1 Aging, Community and Health Research Unit, School of Nursing, Faculty of Health Sciences, McMaster University, Hamilton, Ontario, Canada, 2 School of Public Health and Health Systems and Schlegel-UW Research Institute for Aging, University of Waterloo, Waterloo, Ontario, Canada, 3 School of Human Kinetics, Laurentian University, Sudbury, Ontario, Canada, 4 Northern Ontario School of Medicine and Health Sciences North Research Institute, Sudbury, Ontario, Canada, 5 Centre for Rural and Northern Health Research, Laurentian University, Sudbury, Ontario, Canada

ᴑ These authors contributed equally to this work.
‡ These authors also contributed equally to this work.
* mreid@mcmaster.ca

**Data Availability Statement:** The study data cannot be publicly shared due to ethical restrictions

## Abstract

### Objective

To evaluate the effectiveness of a nurse-led hospital-to-home transitional care intervention versus usual care on mental functioning (primary outcome), physical functioning, depressive symptoms, anxiety, perceived social support, patient experience, and health service use costs in older adults with multimorbidity ($\geq 2$ comorbidities) and depressive symptoms.

### Design and setting

Pragmatic multi-site randomized controlled trial conducted in three communities in Ontario, Canada. Participants were allocated into two groups of intervention and usual care (control).

### Participants

127 older adults ($\geq 65$ years) discharged from hospital to the community with multimorbidity and depressive symptoms.

### Intervention

This evidence-based, patient-centred intervention consisted of individually tailored care delivery by a Registered Nurse comprising in-home visits, telephone follow-up and system navigation support over 6-months.

involving potentially identifying information in accordance with the Hamilton Integrated Research Ethics Board (HiREB) for Hamilton Health Sciences and McMaster University's Faculty of Health Sciences. The participant consent form does not address open public access to the data. Data are available upon request from McMaster University, Faculty of Health Sciences, School of Nursing for researchers who meet the criteria for access to confidential data pending approval from the Hamilton Health Sciences Integrated Research Board. For inquiries, please contact: Dr. Michael McGillion, Associate Professor and Assistant Dean, Research, School of Nursing, Faculty of Health Sciences, McMaster University, Email: mmcgill@mcmaster.ca; Phone: 905-525-9140 x 20275.

**Funding:** Maureen Markle-Reid and Carrie McAiney received funding from the Ontario SPOR Support Unit IMPACT award (grant no: 60502). Ruta Valaitis, Rebecca Ganann, Maureen Markle-Reid, and Carrie McAiney received funding from the Labarge Foundation, McMaster University. This research was also undertaken, in part, thanks to the funding from Dr. Markle-Reid's Tier 2 CIHR Canada Research Chair.

**Competing interests:** The authors have declared that no competing interests exist.

## Outcome measures

The primary outcome was the change in mental functioning, from baseline to 6-months. Secondary outcomes were the change in physical functioning, depressive symptoms, anxiety, perceived social support, patient experience, and health service use cost, from baseline to 6-months. Intention-to-treat analysis was performed using ANCOVA modeling.

## Results

Of 127 enrolled participants (63-intervention, 64-control), 85% had six or more chronic conditions. 28 participants were lost to follow-up, leaving 99 (47 -intervention, 52-control) participants for the complete case analysis. No significant group differences were seen for the baseline to six-month change in mental functioning or other secondary outcomes. Older adults in the intervention group reported receiving more information about health and social services (p = 0.03) compared with the usual care group.

## Conclusions

Although no significant group differences were seen for the primary or secondary outcomes, the intervention resulted in improvements in one aspect of patient experience (information about health and social services). The study sample fell below the target sample (enrolled 127, targeted 216), which can account for the non-significant findings. Further research on the impact of the intervention and factors that contribute to the results is recommended.

## Trial registration

clinicaltrials.gov Identifier: NCT03157999.

## Introduction

As the population of older adults ($\geq$ 65 years) increases, so too does the number of individuals living with multimorbidity, defined as the co-existence of 2+ (or 3+) chronic conditions in the same person [1]. Worldwide, more than half of older adults have multimorbidity [2], with a mean of five chronic conditions per person [3]. High prevalence rates of multimorbidity have been reported in older adults in Canada (43%) [4], the United States (63%) [5], and the United Kingdom (67%) [6], with a significant increase over the last three decades [7, 8]. The increasing prevalence of multimorbidity is driven by a growing aging population as well as a rise in global life expectancies [9–11]. The global population of older adults is expected to reach approximately 16% in 2050 compared to 9% in 2018 [12]. Of the chronic conditions that co-exist, depression is the single most common condition in older adults [13]. Older adults with multimorbidity are two to three times more likely to have depression compared to those without multimorbidity [14]. Depression is a common and serious problem in its own right that is estimated to affect up to 40% of community-living older adults [15]. Although depression can be successfully treated with antidepressant medications or psychosocial interventions, few older adults receive adequate treatment. Untreated or under-treated depression in older adults is associated with a range of negative outcomes, including reduced quality of life, impaired functional status [16], increased use of healthcare services [17], premature admission to long-term care [18], increased mortality [18, 19], and increased burden on family caregivers [1]. As

the number of comorbid chronic conditions increases so too does the likelihood of depression and other mental health conditions [6, 18, 20].

Older adults ($\geq$ 65 years) with multimorbidity and depressive symptoms are arguably the most vulnerable of patient groups. Compared with the general population of older adults, older adults with multimorbidity and depressive symptoms have significantly higher rates of hospital, emergency room, and physician use and costs, and experience frequent transitions between hospital and home [21, 22]. Multiple studies suggest that hospital-to-home care transitions for this population are fragmented and poorly coordinated, resulting in increased hospital readmission rates, adverse medical events, decreased patient satisfaction and safety, and increased caregiver burden [23–31]. Studies in Canada, the USA, and elsewhere have attributed these adverse outcomes to factors such as lack of patient knowledge about available community-based services resulting in suboptimal or delayed utilization of these services [31, 32], conflicting plans of care and instructions from different providers [31, 33–36], medication errors [29–31, 37, 38], lack of timely follow-up with specialists and family physicians after hospital discharge [30, 31, 39], limited engagement of older adults and caregivers in care decisions [29, 40] and preparation for self-care [30, 37, 38, 41–43], lack of support for family caregivers, poor communication and collaboration among providers within and across settings [29, 30, 44], lack of timely and adequate home-based support after hospital discharge [29, 30], untreated or under-treated depressive symptoms [29, 45–47], inadequate community mental health supports [29], and having other unaddressed social and psychological needs during previous hospitalization [30]. Factors related to the social determinants of health, such as lower socioeconomic status, living in rural or remote communities, inadequate housing conditions [30], lack of social support [30], or identifying as ethnocultural minorities, can further exacerbate the challenges associated with hospital-to-home transitions for older adults with multimorbidity and depressive symptoms [48]. Two recent studies conducted in Ontario reported that as many as 44% of patients in Ontario do not attend recommended post-discharge appointments for follow-up care after hospital discharge because of issues such as low health literacy, financial concerns, and a lack of social supports [39, 49].

Transitional care interventions have been recommended to enhance coordination and continuity of health care in community-living older adults with complex needs transitioning from hospital to home, and have been linked to decreasing hospital readmissions and other positive outcomes [31, 44, 50–56]. However, these trials have been largely based on transitional care interventions for single conditions, which often excluded patients with multimorbidity or depressive symptoms [57–61]. As a result, the effectiveness of these interventions for older adults with multimorbidity and depressive symptoms is undetermined [62–64]. Key components of effective hospital-to-home transitional care interventions have been identified, including a comprehensive assessment of patients' current and evolving health care and support needs; patient, family, and caregiver involvement in transition planning; patient, family, and caregiver education support and training; care coordination and system navigation; medication review and support; coordinated team-based care; holistic, person-centred care; and developing individualized care plans [31, 61, 65]. However, what elements should be included in transitional care interventions for older adults with stroke and multimorbidity and depressive symptoms remains inconclusive.

This team designed and tested a new hospital-to-home transitional care intervention to address this gap in knowledge. The Community Assets Supporting Transitions (CAST) is a 6-month, tailored, nurse-led, patient-and caregiver-centred intervention designed to support families and caregivers and foster collaboration between primary care and other interdisciplinary service providers, both within and outside of the health sector, in delivering home and community services. The intervention was adapted from a previous nurse-led mental health

promotion intervention for older home care clients with multimorbidity and depressive symptoms [50]. While this intervention did not specifically focus on transitional care, 60% of the older adult study participants reported one or more hospital admissions in the last six months, thus serving as a strong foundation for the hospital-to-home transitional care intervention tested in this trial. The intervention also included most of the key elements recommended in best practice guidelines for transitional care [31, 40, 65–68]. The results of our feasibility study of the effectiveness of this intervention showed that the intervention was feasible to implement, and resulted in a statistically significant reduction in depressive symptoms, significant improvements in mental and physical functioning, and a statistically significant reduction in the use of hospitalization, ambulance service utilization and emergency room visits [50].

## Objectives

Primary Objective: To compare the effect of a 6-month transitional care intervention versus usual care on the primary outcome–mental functioning–in older adults with depressive symptoms and multimorbidity transitioning from hospital-to-home.

Secondary Objective: To compare the effect of a 6-month transitional care intervention versus usual care on secondary outcomes–physical functioning, depressive symptoms, anxiety, perceived social support, patient experience, and healthcare costs–in older adults with depressive symptoms and multimorbidity transitioning from hospital-to-home.

## Methods

A multi-site, pragmatic, randomized controlled trial was conducted in Ontario, Canada (ClinicalTrials.gov: NCT03157999). The study was designed to be highly pragmatic, using the criteria described in the Pragmatic Explanatory Continuum Indicator Summary-2 tool [69, 70]. Pragmatic features included the recruitment of participants representative of the population presenting in the hospital setting, the flexible delivery of the intervention by Registered Nurses (RNs) from the setting, the use of patient-relevant outcomes (e.g., quality of life, patient experience), and the use of intention-to-treat analysis. Details of the study design and outcome measures are reported in the published study protocol [71]. The methods, results, and flow of participants through the study are presented here according to the CONSORT statement for pragmatic randomized controlled trials [70]. The key design features are summarized below.

### Participants and recruitment

Participants were recruited from three large academic hospitals within three geographical areas in Ontario, Canada. These three communities were selected for their diversity with respect to geography (e.g., rural, urban), socio-economic, and language (i.e., English, French) characteristics. Study recruitment was conducted during 2017–2018 and spanned 5–11 months, depending on the site. Inpatient older adults were screened prior to hospital discharge (n = 825) by a trained recruiter at each hospital for potential inclusion and were eligible to participate if they met the following criteria: 1) aged 65 years or older; 2) planned for discharged from hospital to the community (not long-term care); 3) self-reported at least two chronic conditions; 4) screened positive for depressive symptoms as assessed by a two-item version of the Patient Health Questionnaire (PHQ-2) [72]; 5) not planning to leave the community during the 6-month study period; 6) passed a cognitive screening test (achieved at least 5 correct responses on the Short Portable Mental Status Questionnaire (SPMSQ) [73]; and 7) were competent in English or had an interpreter available who was competent in the English language (this also included French-language in the site with a large Francophone population).

The purpose of the PHQ-2 is not to establish definitively the presence of a depressive disorder, but rather to screen for depressive symptoms as a "first step" approach [72]. Thus, eligible and consenting participants were further evaluated with the Center for Epidemiologic Studies Depression Scale 10-item tool (CESD-10) [74] to determine whether they met the criteria for a depressive disorder (CES-D $\geq$ 10). A trained Research Assistant (RA) contacted potential participants following discharge from hospital to arrange an in-home interview. The RA obtained written informed consent prior to conducting the baseline in-home interview.

## Randomization

Within each study region, participants were assigned to either the intervention or the usual care group following the collection of baseline data, using permuted block randomization administered by a centralized web-based software program (RedCap) that ensured concealment of the allocation from the research team. Participants were allocated to the two groups using a 1:1 ratio and in accordance with the sequence determined by RedCap.

## Intervention

Details regarding the CAST intervention are described in the published study protocol [71]. The intervention was developed using the Medical Research Council Framework for developing complex interventions, which highlights the importance of theoretical and empirical evidence [75]. The intervention was based on Bandura's Social Cognitive Theory [76], where the aim is to build self-efficacy to improve self-management of health conditions and associated risk factors [76], and on research evidence [31, 40, 66–68, 77–82], More importantly, the intervention incorporated input from a range of stakeholders, including patients, providers, and decision-makers from local and provincial health authorities. The stakeholders worked as a team to identify gaps in the delivery of hospital-to-home transitional care which, in turn, informed the core components of the intervention and how they were delivered and tailored in the study settings. The involvement of multiple provider agencies was critical to designing the intervention to ensure that all viewpoints were considered.

The description of the intervention follows the Template for Intervention Description and Replication guidelines [83]. The intervention consisted of usual care plus a 6-month tailored patient- centred intervention delivered by 4 Registered Nurses (RNs) who functioned as Care Transition Coordinators (CTCs) within each of the study regions. The CTCs were not responsible for usual care. All the CTCs had a baccalaureate degree, one CTC had a master's degree, and all had 2–20 years experience working as an RN in both acute care and community settings.

To support intervention fidelity, the CTCs were provided training by the principal investigators and the research coordinator prior to initiation of the intervention to convey key intervention activities, research study procedures, and underlying theories. A standardized training manual was developed that includes key content pertaining to all aspects of the intervention. Training was adapted to the individual needs of the CTCs and included education and role-playing to enhance skills in problem-solving therapy within the context of multimorbidity. Monthly outreach meetings were conducted to enable the principal investigators and the research coordinator to meet with the CTCs to monitor intervention implementation and strategize to address any challenges [84].

The intervention consisted of up to 6 in-home visits (minimum 2), telephone calls, and system navigation support. The CAST intervention was designed to improve both the quality and experience of hospital-to-home transitions. It was based on a patient-centred model and encapsulates strategies included in effective care transitions interventions [31, 40, 66–68], and

recommended in best practice guidelines for system navigation [77], management of depressive symptoms [31, 78, 79], and prevention and management of multimorbidity [80–82].

Each participant was offered monthly in-home visits by the CTCs that were an average of one hour in duration supplemented by telephone calls for 6 months as part of the CAST intervention. The CTCs main activities during the home visits and telephone calls included: 1) conducting a comprehensive assessment of the health and social care needs of older adult participants using standardized tools; 2) identifying and managing depressive symptoms and multimorbidity in accordance with best practice [31, 78–81]; 3) conducting medication review and reconciliation and supporting antidepressant medication management using best practice [85]; 4) providing problem-solving therapy with participants and caregivers using Nezu et al.'s manual [86]; 5) implementing social and behavioural activation strategies tailored to individual needs; 6) providing education to participants and caregivers; and 7) communicating alerts to primary care providers regarding the presence of depressive symptoms, dementia, delirium, suicidal ideation, or medication issues in participants.

During and between home visits, the CTC provided system navigation support that consisted of: 1) identifying and addressing any risk factors for adverse events, e.g., hospital readmissions; 2) arranging community services such as home care and follow-up health-care appointments; 3) facilitating communication between the patient, their family caregiver, and their health care team; 4) supporting linkages and referrals to relevant health and social service providers; 5) developing an individualized patient-centred plan of care [77]; and 6) identifying health care professionals involved in the participant's circle of care and initiating a plan for regular communication and follow-up with them. Consistent with a pragmatic trial design, the intervention was tailored to patient needs and preferences and the local context. For example, patients could decline any number of home visits, and all participants continued to have access to the programs and services normally offered in their community.

## Patient and public involvement

A key component of this patient-oriented research project was the meaningful engagement in all stages of the research process of diverse patients (including family caregivers) who reflected the population of interest. Patient and caregiver research partners with experience in hospital-to-home transitions or depressive symptoms and multimorbidity were actively involved as members on: 1) a Research Steering Committee to provide input on the design of the trial and management oversight, and to inform cross-site implementation of the research; 2) three local Community Advisory Boards to support local implementation of the research at each study site; and 3) the research team as Co-Investigators. Through these structures, patient and caregiver research partners assisted with the identification of the research priorities and questions, selection of patient-relevant outcomes, review of study materials (e.g., consent forms, interview guides), interpretation of study findings, and knowledge dissemination [87]. Patient engagement was grounded in principles of inclusiveness, support, mutual respect, and co-build [88].

## Outcomes and measures

Details regarding outcome measures are described in the study protocol [71]. Outcomes were assessed at baseline and at the 6-month and 12-month follow-up through interviewer-administered questionnaires during a structured in-home interview. The primary study outcome (mental functioning) was measured using the Mental Component Score (MCS) score from the Veterans Rand 12-item health survey (VR-12), a reliable and valid patient-reported outcome measure of quality of life [89, 90]. This outcome was consistent with the overall goal of our

intervention [90], to enhance the mental functioning of older adults with depressive symptoms and multimorbidity. Secondary outcome measures included physical functioning measured using the Physical Component Score (PCS) score from the VR-12 [90]; depressive symptoms measured using the Center for Epidemiologic Studies Depression Scale 10-item tool (CESD-10) [74, 91], anxiety measured using the Generalized Anxiety Disorder 7-item scale (GAD-7) [92, 93], perceived social support measured using the Personal Resource Questionnaire (PRQ-2000) Part 2 [94–96], and patient experience measured using one question from the Client Centred Care Questionnaire (CCCQ) [97], and the complete Intermediate Care for Older People Home-Based-Integrated Care Patient-Reported Experience Measure (IC-PREM) [98]. These instruments have demonstrated reliability and validity and have been used in our previous trials involving community-living older adults with multimorbidity.

Healthcare use was measured using the Health and Social Services Utilization Inventory (HSSUI) [99], which is a reliable and valid self-report questionnaire that measures the use of health and social services [100, 101]. The HSSUI captures use of primary care, emergency department and specialists, hospital days, other health and social professionals, prescribed medications, and lab services. The cost analysis applied unit costs to the service volumes reported in the HSSUI [102] and assumed a societal perspective to inform the broad allocation of resources in the public interest [103].

Guidelines are available for judging clinical significance for the VR-12, but not the other outcome measures. VR-12 developers suggest a minimally important difference (MID) of 3 for interpreting group mean summary score differences (PCS, MCS) [104]. A recent systematic review of RCTs reporting non-significant results emphasized the importance of interpreting confidence intervals in relation to the MID to distinguish "negative" findings from "inconclusive" ones [105]. We applied this recommendation to our study for the PCS and MCS of the VR-12, which have MIDs.

## Blinding

To reduce bias, study participants were blinded to their group allocation (usual care, intervention) and the research assistants who collected the assessment data and statistician who analyzed the data were also blinded. Upon completion of the study at 12-months, participants received a mailed debriefing letter describing the two groups and their group allocation. Usual care providers were also unaware of the participant's group allocation.

## Sample size

The target sample size of 216 (72 from each of the three sites) was calculated for the primary outcome—MCS score of the VR-12 [106]. The calculation assumed 80% power, 2-tailed alpha of 5%, 20% attrition, and a mean (standard deviation) MCS score difference of 6.5 (15.0) as observed in the feasibility study [50]. Using these assumptions, the sample size was 108 (each) for the intervention and control groups.

## Statistical analysis

The reporting of this trial follows the CONSORT guideline for pragmatic RCTs [70]. Descriptive statistics were used to summarize outcome values at baseline, 6 months, and 12 months. Means and standard deviations were used for continuous outcomes, and frequency and percentages were used for categorical outcomes. Analysis of covariance (ANCOVA) was used to assess group differences in the change in primary and secondary outcomes from baseline to 6-months (T2) to determine if the intervention was effective over the 6-month intervention period. The ANCOVA model used the 6-month outcome value as the dependent variable, the

group indicator as the independent variable, and the baseline outcome value as the covariate. Model results were expressed as mean group differences with accompanying 95% confidence limits. Quantile regression was used to examine the group differences in the change in outcomes from baseline to 6-months across quantiles. This method allows us to relax the common regression slope assumption to explore group differences across the distribution of the dependent variable rather than only at the mean [107, 108]. Assessment of group differences in continuous outcomes from 6-months ($T_2$) to 12-months ($T_3$) were assessed using ANCOVA if statistically significant group differences were achieved in the outcomes from baseline to 6-months, with the purpose being to assess the sustainability of the intervention effects.

Z tests of proportions and McNemar tests will be used to assess changes within each group in the number of participants with acute care episodes and the number of acute care episodes from baseline to $T_2$, and from $T_2$ to $T_3$ (if statistically significant group differences were achieved in these outcomes from baseline to 6-months). Acute care episodes included emergency department visits and hospital admissions. Due to the highly skewed nature of cost data, a non-parametric test was used to compare the change in health and social service costs from baseline to $T_2$ for the two groups. Outliers are particularly common for certain healthcare costs such as hospital admissions, thus we conducted a sensitivity analysis to assess the impact of outliers on the cost comparison.

Subgroup analyses will be conducted for the VR-12 outcomes to determine if the intervention was effective for subgroups of participants from baseline to T2 if the overall trial results achieved statistical significance. The following baseline variables were selected a priori for testing subgroup effects: age, sex, number of chronic conditions, depressive symptoms (CESD-10), study site, and dose of the intervention (number of home visits).

Intention-to-treat analysis was employed. Group differences were examined using both a complete case analysis (n = 99) (participants with a complete record at baseline and six-months), and multiple imputation to address missing data for the primary and secondary outcomes. Joint multiple imputation methods were employed as recommended for small samples with an arbitrary missings pattern [109], and five imputations were conducted and pooled to obtain overall parameter estimates and associated confidence intervals. All data analyses assumed a two-sided alpha of 5% and were performed using SAS Version 9.4.

## Ethics approval and consent to participate

The study was conducted in accordance with the Tri-Council Policy Statement, Ethical Conduct for Research Involving Humans [110]. Institutional ethics approval was obtained from: the McMaster University Hamilton Integrated Research Ethics Board (REB) (# 2586); the Office of Research Ethics at the University of Waterloo (#40867); the Laurentian University REB (#6009840), and the REBs from the study sites (Health Sciences North REB # 17–007; Joseph Brant Hospital REB #000-039-17), and renewed yearly as required. Operational approval to conduct the study was obtained from each hospital site. Written informed consent was obtained from all participants by the RA before study enrolment.

## Results

### Study site characteristics

Table 1 provides information related to the characteristics of the regional health authorities within each of the study sides. Sites 1 and 2 served suburban/rural geographies. Site 3 served an urban geography. All the sites had a higher proportion of older adults than the provincial average. Site 1 had the highest turnover of intervention nurses compared to the other two sites.

**Table 1. Study site characteristics.**

| Characteristic | Site 1: Sudbury, Ontario | Site 2: Burlington, Ontario | Site 3: Hamilton, Ontario | Ontario |
|---|---|---|---|---|
| Geographic Density[1] | Suburban/Rural | Suburban/Rural | Urban | |
| Participants Enrolled | 18 | 36 | 73 | |
| Intervention Nurse Turnover | 2 RNs replaced 4 months into intervention | 1 RN added 8 months into the intervention | 1 RN added 12 months into the intervention | |
| Population | ~165,000 | ~205,000 | ~580,000 | |
| Language | 25% French first language; 39% bilingual (English and French) | 81% English as first language | 23.1% non-official language as first language; 73.5% English as first language | 2.4% French first-language; 26.7% non-official language as first language; 66.9% English as first language |
| Ethnocultural Diversity | 3.8% visible minority. 12.5% FNIM | 9.7% visible minority. 0.7% FNIM | 19.0% visible minority. 2.1% FNIM | 22.3% visible minority. 6.2% FNIM |
| Proportion of older adults | 20.4% | 19.3% | 24.5% | 16.4% |
| Proportion of older adults with low income | 2.3% | 4.3% | 7.4% | 5.1% |

RN: Registered Nurse, FNIM: First Nations, Inuit, or Metis

[1] In Canada, the Organisation for Economic Co-operation and Development defines a predominantly rural region as having more than 50% of the population living in rural communities where a rural community" has a population density less than 150 people per square kilometre. Predominantly urban regions have less than 15 percent of their population living in a rural community.

## Eligibility rate

Recruitment ranged from 5 to 11 months depending on the site; significant recruitment challenges were experienced (See discussion). The CONSORT diagram summarizes recruitment, participation, and analysis (Fig 1 and S1 Table). A total of 825 older adults were potentially eligible because they were older adults with a planned discharge to the community and had multimorbidity. Of these older adults, 56% (458/825) were screened for the study and met the remaining eligibility criteria. The most common reason for ineligibility was that potential participants did not screen positive for depressive symptoms (241/367, 66%).

## Enrolment rate

In total, 28% (127/458) of eligible older adults consented and entered the study. Among the eligible participants, almost half refused participation in the study (223/458, 49%); another 18% were either unable to be contacted (42/458, 9%) or had moved out of the study region or to long-term care following hospital discharge (41/458, 9%). Lack of interest in having services in their homes, and participating in a research study, coupled with lack of perceived need for services, were the most common reasons for refusal.

## Attrition rate

Of the 127 enrolled participants, 99 (78%) successfully completed the six-month follow-up. A total of 28 participants were lost to follow-up at six months, yielding an attrition rate of 22%. Of the 127 enrolled participants, 78 (61%) completed the one-year follow-up interviews. A total of 49 participants were lost to follow-up at one year, yielding an attrition rate of 39%. Reasons for loss to follow-up at six months and one year are shown in Fig 1. Attrition in this study was related to several factors. First, 17 (35%) of participants died over the course of the study, 17 (35%) were unable to be contacted, 7 (15%) were hospitalized, and the remainder of those for whom a reason is known were too unwell to participate.

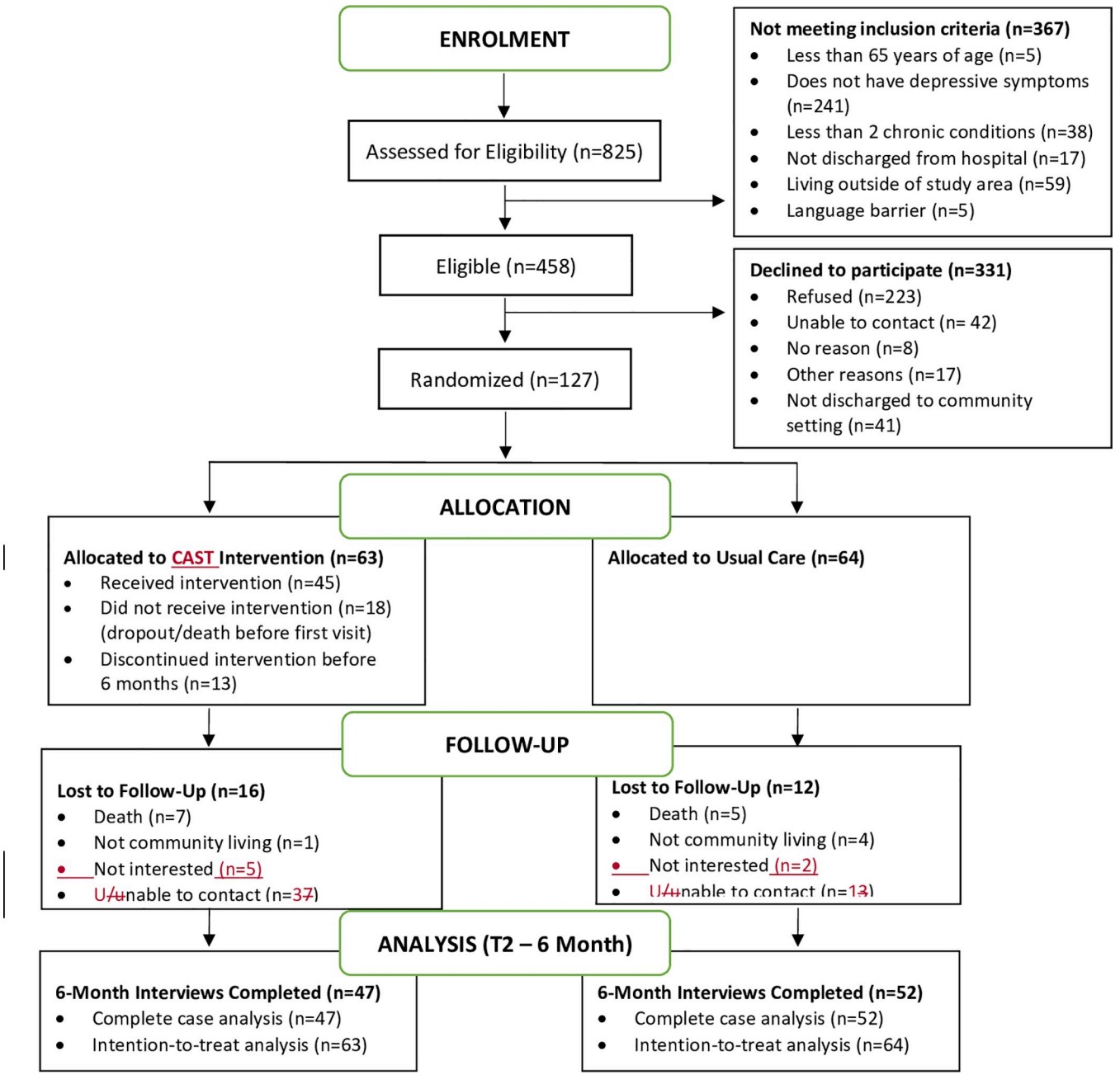

**Fig 1. Study flow diagram.**

## Comparison between dropouts and completers

The baseline characteristics of participants who completed the six-month follow-up (n = 99) were compared to those who dropped out of the study prior to the six-month follow-up (n = 28). Compared to completers, more dropouts lived in a retirement home or supportive living environment (39.3% vs. 12%), reported a history of depression (50% vs. 29.3%), and had lower scores on the VR: MCS-12 (40.7 vs. 43.7). There was no difference between dropouts and participants who completed the six-month follow-up on any other baseline characteristics.

## Baseline characteristics of participants

Baseline characteristics of the participants who completed the six-month follow-up (n = 99) are reported in Table 2. For both groups, approximately 63% of participants were female, 41% were living with a spouse or other family member, 65% were 75 years or older (average of 77 years), 41% were married, and 59% were widowed/divorced. About two-thirds (60%) of the participants had annual incomes of less than CAD$40,000, almost one-third (30%) had less than a high school education, and 38% lived alone. Participants reported VR: PCS-12 and MCS scores at baseline that were significantly lower than published norms for the Canadian population, indicating poor physical and mental functioning [111]. The majority (84.9%) had six or more chronic conditions, with a mean of 8 chronic conditions, and were taking a mean of 8 prescription medications daily. Most participants (79%) self-reported hypertension, and 72% reported arthritis. Depressive symptoms ($\geq$ 10 on CES-D-10) were found in 72% of participants in both groups. For both groups, about 30% of participants self-reported a history of depression, and 31% reported taking an antidepressant. Anxiety symptoms ($\geq$ 5 on GAD-7) were found in 44% of participants.

A higher proportion of participants in the intervention group reported a history of depression (36.2% vs. 23.1%) and took antidepressant medications (25.5% vs. 21.2%) compared with the usual care group. Participants in the intervention group also had lower scores on the VR: PCS-12 (22.2 vs. 26.2) compared with the usual care group.

## Intervention dose

Of 63 intervention participants enrolled at baseline, 45 (71%) received at least one home visit by the CTC. Reasons for not receiving the intervention (all or in part) are shown in Fig 1. Most of the participants who refused the home visits cited that they were not interested in the study (78%). Thirteen additional participants discontinued the intervention early. Over 6 months, the participants received an average of three home visits by the CTC (offered 6). The average duration of the home visits was one hour. For the 45 participants who received at least one home visit, care coordination and system navigation support were cited as the most frequently delivered activity by the CTC nurse (average 4.2 times per participant). This was followed by providing practical and emotional support (average 2.9 times), clinical assessment and screening (average 2.8 times), self-management support (average 2.2 times), behavioural change support (average 2.4 times), health education (average 2 times), health promotion activities (average 1.4 times), and caregiver support (average 1 time).

## Effects of the intervention

**Health outcomes.** The results of the complete case (n = 99) ANCOVA from baseline to 6-months are provided in Table 3. For the primary outcome (VR: MCS-12), the group difference in mean 6-month scores adjusted for baseline values was not statistically significant (mean difference: 1.09; 95% CI: -3.24–5.41). For the secondary outcomes, there were no significant group differences between the intervention and control groups on the: VR: PCS-12 (mean difference: -1.45; 95% CI: -4.96, 2.07), PRQ-2000 (mean difference: 2.95; 95% CI: -1.93–7.83), GAD-7 (mean difference: 1.34; 95% CI: -0.25–2.92) or CES-D-10 (mean difference: 0.80; 95% CI: -1.43–3.03). Multiple imputation results were consistent with the complete case findings. Subgroup analyses were not done since statistical significance was not achieved (see Statistical Analysis above).

Fig 2 provides a graphic interpretation of the MCS and PCS findings. For the MCS, the findings are inconclusive with either usual care or the intervention being potentially superior, since the CI crosses 0 with the upper CI (favours the intervention) and the lower CI (favours

**Table 2. Baseline characteristics of older adults with multimorbidity and depressive symptoms (n = 99) [a].**

| Characteristic | Total | Intervention Group (n = 47) | Usual Care Group (n = 52) |
|---|---|---|---|
| **Sex, n (%)** | | | |
| Male | 37 (37.4) | 18 (38.3) | 19 (36.5) |
| Female | 62 (62.6) | 29 (61.7) | 33 (63.5) |
| **Age in years, n (%)** | | | |
| 65–69 | 13 (13.3) | 7 (14.9) | 6 (11.8) |
| 70–74 | 21 (21.4) | 10 (21.3) | 11 (21.6) |
| ≥ 75 | 64 (65.3) | 30 (63.8) | 34 (66.7) |
| **Type of Accommodation, n (%)** | | | |
| House or Apartment | 87 (87.9) | 43 (91.5) | 44 (84.6) |
| Retirement Home | 12 (12.1) | 4 (8.5) | 8 (15.4) |
| **Marital Status, n (%)** | | | |
| Married, living together | 41 (41.4) | 21 (44.7) | 20 (38.5) |
| Separated, Divorced, Widowed | 58 (58.6) | 26 (55.3) | 32 (61.5) |
| **Education, n (%)** | | | |
| < High School | 29 (29.6) | 11 (23.9) | 18 (34.6) |
| High School | 32 (32.7) | 14 (30.4) | 18 (34.6) |
| Post-Secondary | 37 (37.8) | 21 (45.7) | 16 (30.8) |
| **Annual Income in CAD, n (%)** | | | |
| < $40,000 | 47 (59.5) | 17 (51.5) | 30 (65.2) |
| ≥ $40,000 | 32 (40.5) | 16 (48.5) | 16 (34.8) |
| **Living Arrangement, n (%)** | | | |
| Live Alone | 37 (37.8) | 17 (36.2) | 20 (39.2) |
| Live with Others | 61 (60.7) | 30 (63.8) | 31 (60.8) |
| **Number of Chronic Conditions, n (%)** | | | |
| 0–5 | 15 (15.2) | 5 (10.6) | 10 (19.2) |
| 6 to 10 | 69 (69.7) | 34 (72.3) | 35 (67.3) |
| ≥ 11 | 15 (15.2) | 8 (17.0) | 7 (13.5) |
| **Type of Chronic Conditions, n (%)** | | | |
| Hypertension | 78 (78.8) | 34 (72.3) | 44 (84.6) |
| Arthritis | 71 (72.0) | 36 (76.6) | 35 (67.3) |
| **Anxiety Symptoms, n (%)** | | | |
| ≥ 5 (GAD-7) | 44 (44.4) | 19 (40.4) | 25 (48.1) |
| < 5 (GAD-7) | 55 (55.6) | 28 (59.6) | 27 (51.9) |
| **Depressive Symptoms, n (%)** | | | |
| ≥ 10 (CESD-10) | 70 (72.1) | 32 (68.1) | 38 (76.0) |
| < 10 (CESD-10) | 27 (27.8) | 15 (31.9) | 12 (24.0) |
| **History of Depression, n (%)** | | | |
| Yes | 29 (29.3) | 17 (36.2) | 12 (23.1) |
| No | 70 (70.7) | 30 (63.8) | 40 (76.9) |
| **Antidepressant Medication Use, n (%)** | | | |
| Yes | 31 (31.3) | 20 (25.5) | 11 (21.2) |
| No | 68 (68.7) | 27 (74.5) | 41 (78.9) |
| **Number of Prescription Medications, n (%)** | | | |
| 0–3 medications | 7 (7.2) | 4 (8.7) | 3 (5.9) |
| 4–7 medications | 22 (22.7) | 9 (19.6) | 13 (25.5) |
| ≥ 8 medications | 68 (70.1) | 33 (71.8) | 35 (68.6) |
| Social support[b], mean (SD) | 83.54 (12.9) | 82.29 (15.2) | 83.84 (10.7) |

(*Continued*)

**Table 2.** (Continued)

| Characteristic | Total | Intervention Group (n = 47) | Usual Care Group (n = 52) |
|---|---|---|---|
| Depressive Symptoms[c], mean (SD) | 11.0 (6.60) | 10.6 (6.75) | 11.97 (6.71) |
| Anxiety Symptoms[d], mean (SD) | 5.69 (5.71) | 5.28 (5.14) | 6.14 (6.35) |
| Physical Functioning[e], mean (SD) | 23.91 (10.97) | 22.23 (10.24) | 26.16 (11.61) |
| Mental Functioning, mean (SD) | 43.67 (12.76) | 43.52 (11.91) | 43.98 (14.23) |
| Number of hospital admissions, last 6 months, mean (SD) | 1.55 (0.94); Range: 0–4 | 1.64 (1.03); Range: 0–4 | 1.46 (0.86); Range: 0–3 |

[a] Significance tests were independent t-tests except for number of hospital admissions in 6 months (used non-parametric test–Mann Whitney U–reported z-score & associated p-value).

[b] Measured by Personal Resource Questionnaire (PRQ 2000), scale range 15–105.

[c] Measured by Centre for Epidemiologic Studies Depression 10-item Scale (CES-D-10), scale range 0–30.

[d] Measured by Generalized Anxiety Disorder 7-item Scale (GAD-7), scale range 0–21.

[e] Measured by Physical Component Score (PCS) of the Veterans Rand 12-item Scale (VR-12), scale range 0–100.

[f] Measured by the Mental Component Score (MCS) of the Veterans Rand 12-item Scale (VR-12), scale range 0–100.

the control) exceeding the MID of 3. For the PCS, the findings are inconclusive regarding the superiority of usual care but rules out the superiority of the intervention, since the CI crosses 0 with the lower CI (favours usual care) exceeding the MID of 3 and the upper CI (favours the intervention) does not reach the MID. The results of the quantile regression analyses showed that the intervention consistently outperformed usual care across most of the response range values (baseline to 6-month change) for the PRQ-2000 (perceived social support) (p = 0.03) (Fig 3), although statistical significance was not achieved. ANCOVA from 6-months to one year was not done since statistically significant group differences were not achieved in the outcomes from baseline to 6-months (see Statistical Analysis above).

**Patient experience.**   Chi-square analysis was used to compare the two groups (intervention, control) at baseline and 6 months on participant's care experience, namely one item from the Client Centered Care Questionnaire (CCCQ) [97], and the Intermediate Care for Older People Home-Based-Integrated Care Patient-Reported Experience Measures (IC-PREMs) [98] for the complete cases (n = 99). At 6-months, older adults in the intervention group reported receiving more information about health and social services (p = 0.03) compared with the usual care group. There was no significant group difference between the intervention and control groups on any of the other items on the CCQ or the IC-PREMs Questionnaire (S2 Table).

**Health service use costs.**   The results of the complete case (n = 99) Wilcoxin Rank Sum test from baseline to 6-months are provided in Table 4 for the two groups (intervention, control). The median intervention cost was CAD$449.60 (interquartile range $CAD$0–859.20) per study participant. Despite inclusion of the intervention costs, there was no statistically significant difference between groups in the change in total costs (including or excluding hospital costs) from baseline to 6-months (p = 0.07). For example, cost changes for some services

**Table 3. Group differences in outcomes from baseline to six-months (n = 99).**

| Outcome | Intervention n = 47 | | Control n = 52 | | ANCOVA |
|---|---|---|---|---|---|
| | Baseline Mean (SD) | T2 Mean (SD) | Baseline Mean (SD) | T2 Mean (SD) | Mean Diff (95% CI) [t, p-value] |
| VR: MCS-12 | 43.52 (11.91) | 48.63 (11.62) | 43.98 (14.23) | 47.74 (11.59) | 1.09 (-3.24, 5.41) [0.50, 0.61] |
| PRQ 2000 | 82.29 (15.23) | 83.55 (14.68) | 83.84 (10.71) | 81.59 (13.42) | 2.95 (-1.93, 7.83) [1.2, 0.23] |
| GAD-7 | 5.28 (5.14) | 4.85 (5.28) | 6.14 (6.35) | 3.82 (3.39) | 1.34 (-0.25, 2.92) [1.68, 0.10] |
| CESD-10 | 10.6 (6.75) | 9.82 (7.09) | 11.97 (6.71) | 9.65 (5.25) | 0.80 (-1.43, 3.03) [0.71, 0.48] |
| VR: PCS-12 | 22.23 (10.24) | 24.23 (10.78) | 26.16 (11.61) | 28.07 (10.20) | -1.45 (-4.96, 2.07) [-0.82, 0.42] |

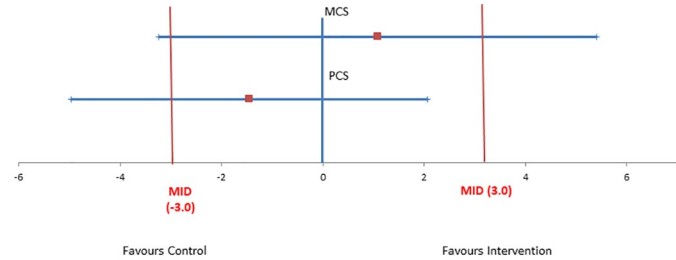

[1] Gewandter, JS et al. Interpretation of CIs in Clinical Trials with non-significant results: systematic review and recommendations. *BMJ Open.* 2017;7:e017288. doi:10.1136/bmjopen-2017-017288

**Fig 2. Interpreting 95% confidence intervals for PCS and MCS ("inconclusive findings").** Note: MID: minimally important difference.

favored the intervention group (family physician and emergency department visits), and others favoured the control group (prescription medications, ambulance, 911 calls, and hospitalization). However, none of these differences were statistically significant. Three extreme outliers were identified (2 in the intervention group and 1 in the control group) that fell well outside the range of health service use. These extreme outliers were excluded from the analysis. The analysis from 6-months to one year was not done since statistically significant group differences were not achieved in the cost of health service use from baseline to 6-months (see Statistical Analysis above).

## Discussion

The purpose of this pragmatic RCT was to test the effects of a six-month hospital-to-home, nurse-led, transitional care intervention for older adults with multimorbidity and depressive

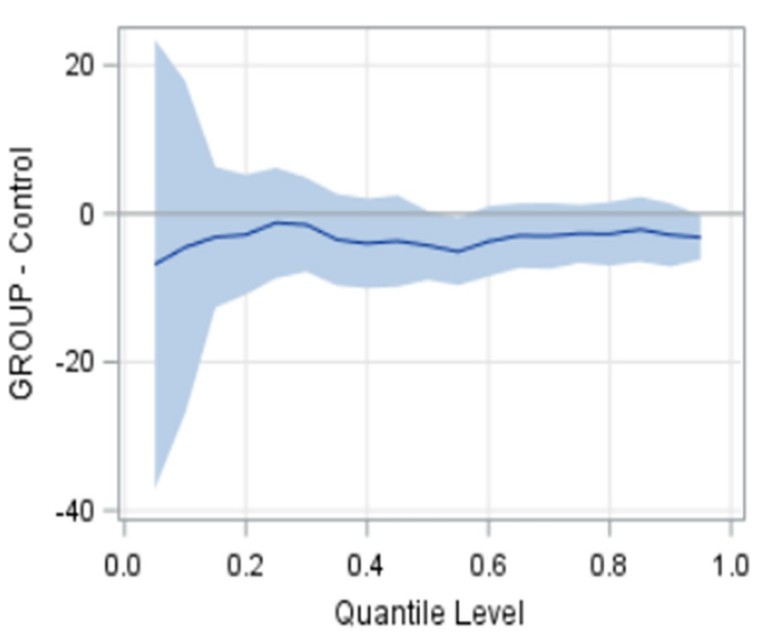

**Fig 3. Quantile regression results: Baseline to six-month change in PRQ.**

**Table 4. Group difference in health and service costs[a] from baseline to six-months (n = 99).**

| Service | Intervention (n = 47) | | Usual Care Group (n = 52) | | Group Differences | |
| --- | --- | --- | --- | --- | --- | --- |
| | Baseline Median (QI, Q3) | 6-Month Median (Q1, Q3) | Baseline Median (QI, Q3) | 6-Month Median (Q1, Q3) | z | p value |
| Family Physician Visits | 348.48 (231.60, 799.92) | 350.64 (198.30, 1178.64) | 231.60 (134.31, 511.60) | 303.94 (143.86, 520.31) | 0.13 | 0.90 |
| Specialist Visits | 195.74 (50.35, 380.16) | 241.23 (50.35, 437.47) | 140.34 (0.00, 351.54) | 164.10 (0.00, 379.43) | 0.25 | 0.81 |
| Home Care | 0.00 (0.00. 122.41) | 0.00 (0.00, 315.85) | 0.00 (0.00, 128.61) | 0.00 (0.00, 160.00) | 0.40 | 0.69 |
| Social & Community | 0.00 (0.00, 0.00) | 0.00 (0.00, 0.00) | 0.00 (0.00, 3.00) | 0.00 (0.00, 0.00) | 0.53 | 0.59 |
| Transportation | 0.00 (0.00, 0.00) | 0.00 (0.00, 24.00) | 0.00 (0.00, 0.00) | 0.00 (0.00, 6.00) | -0.54 | 0.59 |
| Prescription Medications | 711.09 (447.01, 11,977.46) | 803.22 (226.86, 16,006.77) | 783.69 (388.72, 3,797.08) | 507.49 (225.57, 10,266.87) | 1.81 | 0.07 |
| Intervention [c] | 0.00 (0.00, 0.00) | 429.60 (0.00, 859.20) | 0.00 (0.00, 0.00) | 0.00 (0.00, 0.00) | 7.14 | <0.0001 |
| Ambulance & 911 | 264.80 (0.00, 480.00) | 0.00 (0.00, 264.80) | 264.80 (264.80, 480.00) | 0.00 (0.00, 264.80) | 0.88 | 0.28 |
| Emergency Department Visits | 239.31 (239.31, 478.62) | 0.00 (0.00, 478.62) | 239.31 (239.31, 478.62) | 0.00 (0.00, 239.31) | -0.01 | 0.99 |
| Hospitalization | 12,552.00 (6,276.00, 29,811.00) | 0.00 (0.00, 10,983.00) | 15,690.00 (7,845.00, 31,380.00) | 0.00 (0.00, 9,414.00) | 1.29 | 0.20 |
| Total Costs (including Hospital Costs) | 23,369.85 (12,182.52, 44,173.80) | 13,790.06 (2,699.03, 42,379.06) | 24,026.48 (10,420.91, 46,058.39) | 7.354.53 (1,806.41, 28,031.76) | 1.82 | 0.07 |
| Total Costs (excluding Hospital Costs) | 2,172.23 (1,201.49, 15,531.52) | 5.070.81 (2,369.95, 18.367.59) | 1,885.00 (1,303.38, 9,278.66) | 3,074.69 (889,18, 14,427.26) | 1.83 | 0.07 |

[a] Currency CAD

[b] Wilcoxon Rank Sum Test used to determine significance of group differences. p-values are 2-sided.

[c] Includes costs for the intervention group (training, in-home visits). Difference between intervention and control group favours control group.

symptoms on health outcomes (mental and physical functioning, depressive symptoms, anxiety, perceived social support), patient experience, and service use costs. This intervention was based on a patient-centred model, and encapsulated strategies included in effective care transitions interventions [31, 66–68], and recommended in best practice guidelines for management of depressive symptoms [31, 78, 79] and multimorbidity [80–82]. The intervention is well-aligned with health-care reform in Ontario and Canada, which is focused on exploring new health-care models that integrate and coordinate care around the patient and across providers in a way that makes sense for each community and improves patient outcomes [31]. Although the overall benefits of the intervention were inconclusive for mental and physical functioning, the intervention resulted in improvements in one aspect of patient experience, and the potential for significant improvements in perceived social support.

This trial had several strengths. It was designed to be highly pragmatic, using the criteria described in the Pragmatic Explanatory Continuum Indicator Summary-2 tool [69, 70]. As a result, it reflects the effectiveness of the intervention in real-world implementation [112]. Pragmatic features included the recruitment of participants representative of the population presenting in the hospital setting, the flexible delivery of the intervention by RNs from the setting, the use of patient-relevant outcomes, (e.g., quality of life, patient experience), the flexible delivery of the intervention by providers, and the use of intention-to-treat analysis [112]. The baseline rate of depressive symptoms of 72% in the present sample is higher than the 55% rates [113] reported for representative samples of hospitalized older adults at discharge. The use of broad eligibility criteria in a range of diverse study sites with respect to geography (e.g., rural, and urban), socio-economic, and language (e.g., English and French) characteristics enhances external validity. Our study is unique in that it measured the costs of use of a full range of health and social services, from a societal perspective. Previous trials assessing the impact of

transitional care interventions provide little information on costs and most have focused on the cost of institutional care (e.g., hospital, ER, long-term care).

This trial had several limitations that may have contributed to the modest study results. First, only 45 (71%) of intervention group participants received at least one home visit by the nurse. Of that number, 32 (71%) completed the six-month intervention and 13 (29%) withdrew prior to six-months. Generally, this engagement rate is in keeping with other studies of older adults with multimorbidity, such as the 3D trial which reported that reach varied across sites from 38% and 94% (median 66%) [10]. The low intervention engagement rate in our study may have diluted the effectiveness of the intervention. Implementation difficulties and deficiencies are not frequently identified in effectiveness evaluations of complex healthcare delivery interventions [10, 114]. We are currently conducting a process evaluation on the implementation of the intervention to understand the influence of contextual factors on study outcomes and inform decisions about wider implementation of the intervention or the need for further research.

Second, only 28% of eligible participants agreed to participate in the trial (enrolled 127, targeted 216) despite extending the recruitment period of the trial. Generally, this recruitment rate is in keeping with other studies of older adults with multimorbidity, such as our previous nurse-led mental health promotion trial that reported a recruitment rate of 29% [50], the 3D trial that reported a recruitment rate of 33% [10], and the Guided Care cluster trial, where 38% agreed to participate [115]. The challenges recruiting older adults with depression are also well documented [116, 117]. An examination of 114 trials involving older adults with depression conducted between 1994 and 2002 showed that less than one-third (31%) reached their original recruitment targets [118]. Our study sample was well below the target sample size of 216, which may account for the study's non-significant findings and may have led to recruitment bias. The finding that there was no statistically significant difference between groups in per-person health service use costs may not be due to similar cost of the two interventions, but due to a lack of power (small sample size). We invested significant resources and implemented numerous strategies to encourage recruitment that are outlined in detail in the study protocol [71]. However, the results highlight the continued need to identify, test and implement innovative methods to enhance recruitment of this population.

Third, despite randomization, there was some chance imbalance in the groups at baseline. A higher proportion of participants in the intervention group reported a history of depression compared with the usual care group (Table 2). This may have reduced the comparative benefit of the intervention.

Fourth, we should acknowledge the retention challenges encountered in this study. We had attrition rates of 22% at six-months and 39% at one-year. There was a slightly higher loss to follow-up in the intervention group at both six months (25% vs. 19%) and one-year (45% vs. 33%) compared to the control group (see Fig 1). Our retention rate is typical of trials in this population, such as our previous nurse-led mental health promotion trial that reported a retention rate of 61% at one year [50], and the Guided cluster trial where 55% completed the final interview [115]. However, attrition may have resulted in self-selection bias, because the drop-outs differed from those individuals who completed the study in that they were more likely to live in a retirement home or supportive living environment, reported lower levels of mental health functioning, and were more likely to report a history of depression. Based on this, it would appear as if the dropouts were a somewhat lower-functioning group than those who were retained in the study. Future research is warranted to identify effective strategies to recruit and retain vulnerable older adult populations [119]. Finally, the large number of secondary outcomes raises the possibility of false-positive findings due to multiple testing. However, in pragmatic trials, it is important to include a range of outcomes that are relevant to

patients, providers, and decision-makers [112]. A recent consensus study identified a core set of outcomes specifically for multimorbidity research, and this trial includes all of these outcomes (health-related quality of life, mental health outcomes, and mortality) [120].

The finding that the overall benefits of the intervention were inconclusive for mental and physical functioning is consistent with several large multimorbidity trials [10, 61, 115]. This is due in part to the heterogeneity in the type of interventions and the characteristics of the study participants included in these trials [10]. The non-significant findings in our study may also be due, in part, to the limitations and challenges of implementing pragmatic trials. The literature suggests that while the features of pragmatic trials support the applicability of the results to real-world practice, they may also reduce effect sizes. In this study, these features include recruiting and retaining heterogeneous populations, lack of a placebo, and suboptimal delivery of the intervention [121].

Nevertheless, it is possible that our transitional care intervention, which is similar to multimorbidity interventions in terms of the components, such as the 3D trial [10], and the Guided Care cluster trial [115], improves older adults' perceptions of the quality of their care but not the quality of their lives. Older adults in the intervention group reported receiving more information about health and social services (p = 0.03) compared with the usual care group. Information provision regarding available health and social services is a key aspect of patient's care experience in the IC-PREM [98]. In a recent systematic review of the reliability and validity of eighty-eight patient-reported experience measures, Bull et al. [122] reported several other tools that included information provision as a component of measuring patient experience. However, the fact that older adults received more information does not indicate whether the information was appropriate, and whether lack of information was an issue. Nevertheless, improving patient experience is one of the triple aims of health care [123], and improving patient experience and incorporating patient's perspectives into the design and delivery of health services has been shown to improve patient health, healthcare service delivery and quality of care [124]. This is particularly important for older adults with multimorbidity, who are managed by different providers, often through many unconnected care episodes [125]. Nevertheless, providing care that is demonstrably more patient-centred is arguably sufficient justification for implementation, especially since the intervention was found to be cost neutral relative to usual care. Previous trials assessing the impact of transitional care interventions have focused on acute care readmission rates as the primary measure of effect, with limited attention to patient-relevant outcomes, such as patient experience.

Exploratory analysis suggests that there was a consistent trend of greater improvement in the level of perceived social support in the intervention group compared to the usual care group over a broader range of values (beyond the mean). This finding is noteworthy given that lower levels of perceived social support are linked to increased hospitalization rates [126]. However, further research is needed to replicate and confirm this exploratory finding.

## Conclusion

This pragmatic trial of a nurse-led transitional care intervention for older adults with multimorbidity and depressive symptoms transitioning from hospital to home demonstrated inconclusive results for mental and physical functioning (MCS and PCS of VR-12), improvements in one aspect of patient experience, and the potential for significant improvements in perceived social support. Further research on the impact of the intervention and the factors that contribute to the results seen is recommended given the high prevalence of depression among older adults with multimorbidity transitioning from hospital to home, and the low rate of recognition and treatment of depressive symptoms in this high-risk population. Future

research is also needed to identify strategies to improve recruitment and retention rates to ensure adequate sample size; improve the reach of the intervention; and to understand the influence of these contextual factors on study outcomes to inform decisions about wider implementation of the intervention and the need for further research.

## Supporting information

**S1 Table. CONSORT 2010 checklist of information to include when reporting a pragmatic trial.**
(DOCX)

**S2 Table. Group differences in patient experience outcomes at baseline and six-months (CCQ, IC-PREMs) (n = 99).**
(DOCX)

**S1 File. Study protocol approved by the McMaster University Hamilton Integrated Research Ethics Board.**
(DOCX)

## Acknowledgments

We thank the older adults who participated in this study, as well as the Nurse Care Transition Coordinators (CTCs) at the participating communities who provided the intervention. We also thank Hamilton Health Sciences, Joseph Brant Hospital, and Health Sciences North who recruited study participants, one hospital site who supplied and supervised the CTCs who provided the intervention, and the Centre for Rural and Northern Health Research (CRaNHR) who supplied and supervised the RAs. A final thanks to the RAs and the research team in the Aging, Community and Health Research Unit, School of Nursing, McMaster University, Hamilton, Ontario, Canada for supporting this study.

## Author Contributions

**Conceptualization:** Maureen Markle-Reid, Carrie McAiney, Gail Heald-Taylor, Fran McMillan, Penelope Petrie, Jenny Ploeg, Diana J. Urajnik, Carly Whitmore.

**Formal analysis:** Maureen Markle-Reid, Kathryn Fisher.

**Funding acquisition:** Maureen Markle-Reid.

**Investigation:** Maureen Markle-Reid, Carrie McAiney, Kathryn Fisher, Rebecca Ganann.

**Methodology:** Maureen Markle-Reid, Carrie McAiney, Kathryn Fisher, Rebecca Ganann, Jenny Ploeg, Carly Whitmore.

**Project administration:** Maureen Markle-Reid, Carrie McAiney, Alain P. Gauthier, Janet E. McElhaney, Diana J. Urajnik.

**Resources:** Alain P. Gauthier, Janet E. McElhaney, Diana J. Urajnik.

**Supervision:** Maureen Markle-Reid, Carrie McAiney, Kathryn Fisher, Alain P. Gauthier, Diana J. Urajnik.

**Validation:** Gail Heald-Taylor, Fran McMillan, Penelope Petrie.

**Writing – original draft:** Maureen Markle-Reid.

**Writing – review & editing:** Carrie McAiney, Kathryn Fisher, Rebecca Ganann, Alain P. Gauthier, Gail Heald-Taylor, Janet E. McElhaney, Fran McMillan, Penelope Petrie, Jenny Ploeg, Diana J. Urajnik, Carly Whitmore.

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
