## [Decision Letter · Decision Letter 0]

20 Oct 2020

PONE-D-20-18099

Effectiveness of a nurse-led hospital-to-home transitional care intervention for older adults with multimorbidity and depressive symptoms: A pragmatic randomized controlled trial

PLOS ONE

Dear Dr. Markle-Reid,

Thank you for submitting your manuscript to PLOS ONE. After careful consideration, we feel that it has merit but does not fully meet PLOS ONE’s publication criteria as it currently stands. Therefore, we invite you to submit a revised version of the manuscript that addresses the points raised during the review process.

The manuscript has been evaluated by three reviewers, and their comments are available below.

The reviewers have raised a number of concerns that need attention. They request additional information on methodological aspects of the study , revisions to the statistical analyses and to the conclusions reported.

Could you please revise the manuscript to carefully address the concerns raised?

We look forward to receiving your revised manuscript.

Kind regards,

Carmen Melatti

Associate Editor

PLOS ONE

Journal Requirements:

2.We note that you have indicated that data from this study are available upon request. PLOS only allows data to be available upon request if there are legal or ethical restrictions on sharing data publicly. For information on unacceptable data access restrictions, please see http://journals.plos.org/plosone/s/data-availability#loc-unacceptable-data-access-restrictions.

Reviewers' comments:

Reviewer's Responses to Questions

**Comments to the Author**

1. Is the manuscript technically sound, and do the data support the conclusions?

Reviewer #1: Partly

Reviewer #2: Yes

Reviewer #3: Partly

2. Has the statistical analysis been performed appropriately and rigorously? 

Reviewer #1: No

Reviewer #2: Yes

Reviewer #3: Yes

3. Have the authors made all data underlying the findings in their manuscript fully available?

Reviewer #1: Yes

Reviewer #2: Yes

Reviewer #3: Yes

4. Is the manuscript presented in an intelligible fashion and written in standard English?

Reviewer #1: Yes

Reviewer #2: Yes

Reviewer #3: Yes

5. Review Comments to the Author

Reviewer #1: The study aimed to evaluate the effectiveness of a nurse-led hospital-to-home transitional care intervention versus usual care on mental functioning (primary outcome), physical functioning, depressive symptoms, anxiety, perceived social support, patient experience, and health service use costs in older adults with multimorbidity (> 2 comorbidities) and depressive symptoms.

This is quite an interesting study but limited by low sample size. The manuscript can be further improved.

Comments

Method

Randomization

Page 7 Line 179, information on concealment to be stated. The word software to be stated.

Page 8 Line 184-185 & Line 188. the number of CTCs/Registered nurse involved and who provided the training to the CTCs to be stated.

Sample size calculation

Page 11 Line 259-262, 1 or 2 tailed test and the sample size allocated for the two intervention groups after calculation to be stated.

Statistical Analysis

Page 11 Line 272-273, more detail to be provided for the analysis.

Page 11 Line 281-282, data to be presented.

Page 11 Line Line 277 & 287, Z tests of proportions and McNemar tests ‘were’ used & Sub group analysis ‘were’ conducted, respectively to be revised to ‘will be’.

Page 10, the mode administration of the questionnaire whether self-administered or interviewed to be clearly stated in the method section. Any bias arises (if interviewed) to be discussed.

Results

Page 17 Table 2, sample size incorrectly labeled for the group (n=63, n=64). Symbol % for individual figures to be omitted, Based on CONSORT guidelines, all statistical tests for baseline comparison to be avoided. For some figures in the Anxiety, sd is larger than mean. For the number of hospital admission, type of data, minimum and maximum value to be stated.

Page 20 Table 3, the row arrangement for the outcome to be re-arranged and to begin with primary outcome; VR: MCS-12 and followed by secondary outcomes VR:PCS-12 etc. n to be stated for the groups.

Page 19 Line 429-430. for standardization in highlighting the confidence level, ‘to’ to be replaced with ‘dash’

Page 20 Line 435, for easy reference MID value to be stated.

Page 20 Line 440-441, p value to be stated.

Page 22 Table 4, table alignment requires adjustment. Word ‘Stat’ to be omitted from z. Currency CAD to denoted in the table or table footnote. Typo Differences

Complete case analysis to be stated for Table 4 in the text.

Findings of multiple imputation for patient experience and health service use costs to be stated.

Figure 1, the intervention name to be stated, Not interested/Unable to contact to be separated.

Information on whether any participants affected by the study after the intervention to be stated (if any).

Power of study could be tested and discussed based on the final sample size in the study.

Effect size could be presented.

For those data analysis not shown, the data analysis can be attached as supplementary.

Decimal point for percentages to be standardized throughout the manuscript (text/tables).

[ ] to be used to cite references in the text.

Funding support to be written in the acknowledgement.

Reviewer #2: This is a clearly written and through report of a pragmatic RCT. It is evident that the authors carefully documented and discussed the strengths of the study, including the pragmatic approach, the measurement of health and social services costs, and the assessment of depressive symptoms. Limitations were also clearly identified and discussed including the sample size, attrition rate, and differences in baseline characteristics between groups.

My concerns about the manuscript primary relate to CONOSRT checklist items related to sample size (item 7a) and outcomes and estimation (items 17 a/b). In this pragmatic RCT, there were clearly described challenges in obtaining the required sample size to effectively measure the primary outcome. A sample of about 60% of the optimum number was obtained. The lack of significant findings may be attributed to the sample size. The authors included a discussion about the importance of reporting minimally important differences between groups, but it is not clear what the value of the minimally important differences are.

The authors include in the discussion that they are conducting a process evaluation of the implementation of the intervention and it is likely that this will yield important insights into the intervention and the results of the pragmatic RCT.

While the authors present a clear and thoughtful report of a pragmatic RCT, the results appear to be quite inconclusive. The authors could consider examining some of the lessons learned from this research (e.g. measurement of depression, innovative recruitment methods) could be included a brief report versus a report of the RCT as a whole.

Reviewer #3: Thank you for the opportunity to review this manuscript. This paper reports the results of a pragmatic randomize controlled trial on the effectiveness of a nurse-led hospital-to-home transitional care intervention for older adults with multimorbidity and depressive symptoms. This was a well-designed study that addressed an important research gap. However, the authors struggled to recruit and retain a sufficient number of participants, so they faced substantial sample size problems. The authors need to be applauded to openly communicate and discuss these struggles – which is valuable information in and by itself. However, there are a couple of key issues that I think the authors need to address:

Abstract, results: The authors state that having depressive symptoms along with multi-morbidity was an inclusion criterion. How is it possible then that only 56% of the participants had depressive symptoms? What does ‘clinically significant’ mean in this context? Please clarify.

Abstract, results: As per the CONSORT guideline for abstracts of RCTs, the number of participants analyzed in each group needs to be stated. The authors only state how many participants they enrolled but not how many they ended up including in their analyses (by study group). Please add this information. This also raises the question whether the analyses were done per protocol, as treated, intention to treat? Please clarify. There are no details regarding the statistical analyses in the abstract at all. This would probably help clarify my next comment too.

Abstract, results: Please clarify the following statement: “No significant group differences were seen for the baseline to six-month change in mental functioning (mean difference: 1.09, 95% CI: -3.24, 5.41, p=0.61)”. This sentence seems to mix a couple of things. If the authors compared group differences in baseline to follow-up CHANGES, then a simple mean difference is not the appropriate outcome. The way the authors present their results makes me think that they compared the follow up group means between the intervention group and the control group. A group comparison of baseline to follow up changes would require the comparison of the intervention group change (baseline to follow up mean difference in this group) to the control group change (baseline to follow up mean difference in that group). Please be clear about what was done. As per my comment above: a sentence or two related to the analytic approach in the methods section might clarify these issues and help the authors to present their results more clearly. The analyses are nicely described in the methods section of the paper.

Abstract, conclusions: I think the conclusions are too strong. First, according to the results section in the abstract, only one outcome improved – but in the conclusions the authors speak about measures (plural). Second, receiving more information is not necessarily a patient experience measure. Whether patients receive more or less information does not indicate at all whether the information was appropriate and whether lack of information actually was the issue. Most importantly, it does not tell us anything about the patient’s experience. Therefore, I suggest the authors limit their statement to ‘patient information measures’. Stating that the intervention was cost neutral because detected differences were not statistically significant is also way too strong a statement, given that the study was under-powered to even detect differences in the main outcome – due to recruitment challenges. It was very likely not powered either to detect differences in this secondary outcome (costs). Therefore, the reason for statistically non-significant differences in costs may not be due to similar costs of the two interventions, but due to a lack of power (small sample size). Table 4 at least suggests that tendentially the total costs of the intervention exceeded those of usual care.

Abstract: no study registration number available in the abstract (although required according to the CONSORT guideline for abstracts of RCTs).

Introduction: In lines 108 and following, the authors make a case for why this specific intervention they tested is needed. However, I think some more specifics are required here. The authors state that multiple studies are available demonstrating poor quality of ‘care transitions’. I think what the authors refer to are transitions from an acute care hospital to the community – not any of the other possible healthcare transitions (e.g., hospital-rehab, hospital-LTC, community-hospital, etc.). Please clarify and be consistent. More importantly, the study reported here is set in Canada but only one of the three studies cited is a Canadian study, the other two studies are Norwegian. Please elaborate on the concrete issues present and to be addressed in the Canadian context (or more precisely in Ontario where the study was situated). The following arguments about factors contributing to fragmented and poor quality transitions also largely cite US studies, a few are European, and many are more than 10 years old. I am not sure whether these studies are most suitable to inform an intervention study that addresses issues in a Canadian context – at least not without adding evidence that the exact issues and contributing factors that international studies found are present in the Canadian context too. Otherwise, an intervention study addressing these factors may not address the right causes and therefore may not be effective.

Also, in their study protocol, the authors state that they used components that were demonstrated to be effective in previous transitional interventions (e.g., home visits, telephone contacts, care coordination) and they cite studies demonstrating that these previous interventions were effective. If there are effective transitional interventions, why not use an existing one and adapt it to the specific context conditions Ontario, Canada? I think the authors do not sufficiently justify why a new intervention is needed, rather than building on an existing one.

Methods, intervention description: Neither in their trial protocol nor in this results paper do authors give any details on how the intervention was developed – i.e. based on which theoretical foundations, evidence, and methods to develop complex interventions (e.g. the MRC framework, the Behaviour Change Wheel, Intervention Mapping, etc.). Also, from the intervention description it is unclear whether, to what extent and how people with lived experience (patients and their families) and decision makers were involved in developing the intervention. I suggest to move the patient & public involvement section up, right after (or before) the intervention description – if allowed by the author instructions.

Methods, outcome measures: in their study protocol, as well as, in this results paper, the authors merely list the tools used to assess study outcomes – and while references are cited that may include results on the tools’ psychometric properties, the authors do not mention at all for any of the tools used what is known about their reliability and validity. This could be done in an appendix if the authors have to save space, but robust psychometric properties of research tools are a key component of rigour in a trial like this one – and they have to be reported.

Results, baseline characteristics: same comment as in abstract: the authors need to define what ‘clinically significant’ depressive symptoms mean, compared to clinically non-significant symptoms. Also please explain why not everyone had depressive symptoms if having these symptoms was an inclusion criterion. Most importantly, the authors report that 56% of participants had clinically significant depressive symptoms in the abstract while here they report that 72% of participants had clinically significant depressive symptoms. Even more confusing, further down the proportion of participants with clinically significant depressive symptoms is 69%. If the discrepancies between the abstract and the baseline characteristics section are because the abstract reports numbers at follow up, please be clear about that – and definitely clarify the two different proportions of participants with clinically significant depressive symptoms in this section (I think 72% is the number for both study groups taken together, 69% is the proportion in the intervention group).

Discussion, first paragraph: same comments as in abstract – the conclusions are too strong. Neither is it justified to state that patient experience improved (since more information is not a proxy for good patient experience), nor does an under-powered study allow to conclude that the lack of statistical significance suggests that costs between the intervention and usual care did not differ. The same is true for lines 576 and following (more information does not equal better quality of care or better patient experience – more is not necessarily better).

Discussion, lines 501, 502: now the authors state again that the baseline rate of clinically significant depressive symptoms was 56%, which is different from the numbers reported in the results section. In line 510 the number is back to 69%. I think the 69% refers to people with either clinically significant depressive symptoms or anti-depressant use, but please clearly explain and align those numbers throughout the paper.

Discussion, limitations – recruitment/retention challenges: I highly appreciate the efforts and resources the authors invested to recruiting participants and I can only imagine the struggles and frustration encountered. It is highly valuable to publish studies that openly report these challenges so other research teams can be better prepared for the issues they will encounter. It is also highly valuable that the authors are currently preparing a publication on the process evaluation findings of their trial. Systematically assessing what worked and what did not – and possible mitigation strategies – will be an important contribution to the literature. While I understand that the authors want to do more work on these issues and publish those results separately, I would still appreciate if they could add a couple of key learnings to the discussion section – i.e. what would they do differently now and what advice do they have for other research teams who intend to do intervention research with this population.

Conclusions: same comments as to the abstract and the discussion sections – wording needs to be revised so it adequately reflects what can be concluded from this study

6. PLOS authors have the option to publish the peer review history of their article (what does this mean?). If published, this will include your full peer review and any attached files.

Reviewer #1: No

Reviewer #2: No

Reviewer #3: **Yes: **Matthias Hoben

---

## [Author Response · Author response to Decision Letter 0]

26 Nov 2020

Reviewer 1 Feedback 

Methods 

1. Randomization: Page 7 Line 179, information on concealment to be stated. The word software to be stated. Response: We clarified that we used a centralized web-based software program (RedCap) that ensured allocation concealment in the Randomization sub-section of the Methods. This change is now on Page 9 Lines 209-210. 

2. Page 8 Line 184-185 & Line 188. the number of CTCs/Registered nurse involved and who provided the training to the CTCs to be stated. Response: We revised the Intervention sub-section of the Methods to include the number of CTCs/Registered Nurses involved and who provided the training to the CTCs. This change is on Page 10 Line 227. 

3. Sample size calculation: Page 11 Line 259-262, 1 or 2 tailed test and the sample size allocated for the two intervention groups after calculation to be stated. Response: We revised the Sample Size sub-section of the Methods to state that a 2-tailed alpha was used to calculate the sample size and that this results in a sample size of 108 (each) for the intervention and control groups. This change is on Page 13 Lines 321-324. 

4. Statistical Analysis: Page 11 Line 272-273, more detail to be provided for the quantile regression analysis. Response: We revised the Statistical Analysis sub-section of the Methods to add more detail about the quantile regression that was used in the analysis. This change is on Page 14 Lines 334-337. 

5. Page 11 Line 281-282, data on the Risk Difference to be presented. Response: We deleted this from the Statistical Analysis sub-section of the Methods since we did not do this analysis in the study. This change is on Page 14-15 Lines 346-348. 

6. Page 11 Line 277 & 287, Z tests of proportions and McNemar tests ‘were’ used & Subgroup analysis ‘were’ conducted, respectively to be revised to ‘will be’. Response: We revised the wording in the Statistical Analysis sub-section of the Methods. This change is on Page 14 Line 342 and Page 15 Line 352. 

7. Page 10, the mode administration of the questionnaire whether self-administered or interviewed to be clearly stated in the method section. Any bias arises (if interviewed) to be discussed. Response: We revised the Outcomes and Measures sub-section of the Methods to indicate that interviewer-administered questionnaires were used. No bias was associated with the administration of the questionnaires by the interviewer. This change is on Page 12 Lines 283-284. 

Results

8. Page 17 Table 2, sample size incorrectly labeled for the group (n=63, n=64). Symbol % for individual figures to be omitted, Based on CONSORT guidelines, all statistical tests for baseline comparison to be avoided. For some figures in the Anxiety, sd is larger than mean. For the number of hospital admission, type of data, minimum and maximum value to be stated. Response: The sample size was corrected in Table 2. The symbol % for individual figures was omitted. We did not conduct a statistical test for baseline comparison. The SD for anxiety for the total group and the usual care group is larger than the mean. The range or minimum and maximum values for the mean number of hospitalizations were added. This change is on Page 20-21 Table 2. 

9. Page 20 Table 3, the row arrangement for the outcome to be re-arranged and to begin with primary outcome; VR: MCS-12 and followed by secondary outcomes VR:PCS-12 etc. n to be stated for the groups. Response: Table 3 was revised so that the VR: MCS-12 was listed first in the table followed by the secondary outcomes and the n was stated for each of the study groups. This change is on Page 23 Table 3. 

10. Page 19 Line 429-430. for standardization in highlighting the confidence level, ‘to’ to be replaced with ‘dash’. Response: “To” in the confidence level was replaced with a “dash” in the Effects of the Intervention (Health Outcomes): sub-section of the Results. This change is on Page 22 Line 499 and 502-503. 

11. Page 20 Line 435, for easy reference MID value to be stated. Response: The MID value was added to the Effects of the Intervention (Health Outcomes): sub-section of the Results. This change is on Page 23 Line 509 and 511. 

12. Page 20 Line 440-441, p value to be stated. Response: The p-value was added to the Effects of the Intervention (Health Outcomes): sub-section of the Results. This change is on Page 23 Line 514. 

13. Page 22 Table 4, table alignment requires adjustment. Word ‘Stat’ to be omitted from z. Currency CAD to denoted in the table or table footnote. Typo Differences

Complete case analysis to be stated for Table 4 in the text. Response: Table 4 was adjusted, the word “stat” was omitted from z, the current CAD was denoted in a table footnote. The results of the complete case (n=99) from Table 4 was added to the text in the Effects of the Intervention (Health Service Use Costs): sub-section of the Results. These changes are on Page 25-26 Table 4. 

14. Findings of multiple imputation for patient experience and health service use costs to be stated. Response: We clarified in the Statistical Analysis sub-section of the Methods that multiple imputation was only conducted for the primary and secondary outcomes; not for patient experience or health service use This change is on Page 15 Line 359. 

15. Figure 1, the intervention name to be stated, Not interested/Unable to contact to be separated. Response: We added the name of the intervention and separated out the number of participants who were lost to follow-up because they were not interested and the number of participants who were lost to follow-up because they were unable to be contacted in Figure 1. These changes are in the revised Figure I. 

16. Information on whether any participants affected by the study after the intervention to be stated (if any). Response: As indicated in the Statistical Analysis sub-section of the Methods (Page 24 Line 339-341, 344-345), assessment of group differences in continuous outcomes from 6-months (T2) to 12-months (T3) were assessed using ANCOVA if statistically significant group differences were achieved in the outcomes from baseline to 6-months. Because no statistically significant differences were found in the outcomes from baseline to 6-months, we did not conduct an analysis of the group differences after the intervention ended from 6-months to 12-months (T3). 

17. Power of study could be tested and discussed based on the final sample size in the study. Response: We did not include a post-hoc power calculation in the paper because it is our understanding that these are not recommended. They are considered appropriate in relation to planning a future study, but not in relation to the study that has been completed. The problem is, post-hoc power calculations cannot be used to calculate the expected value of the true power of a study. Observed power is not considered to be a useful statistical concept because observed (or post-hoc) power and p-values are directly related: whenever a test is not significant (high p-value), retrospective power at the observed effect sizes must always be low, and conversely, whenever a test is significant (low p value), retrospective power must always be high. Therefore, the post-hoc power calculation does not add new information beyond the p-values that have already been reported. Since there does not appear to be support from the scientific community for the reviewer’s request to conduct a post-hoc power calculation, we prefer to not include these in the paper. Please see the references below for further information on why post-hoc power calculations are not recommended:

(Hoenig, JM & Heisey, DM. (2001). The Abuse of Power: The Pervasive Fallacy of Power Calculations for Data Analysis. The American Statistician. 55(1): 1-6. Available at: https://www.vims.edu/people/hoenig_jm/pubs/hoenig2.pdf)

18. Effect size could be presented. Response: We are reluctant to provide effect sizes resulting from an underpowered study because they would not be reliable/representative. 

19. For those data analysis not shown, the data analysis can be attached as supplementary. Response: We added all the data to the text or the Supplementary files and removed the statement “data not shown” from the paper. These changes are on Page 19 Line 446 and 451, Page 20 Line 459, Page 24 Line 532. 

20. Decimal point for percentages to be standardized throughout the manuscript (text/tables). Response: We standardized the decimal points for percentages throughout the manuscript in the text and the tables. 

21. [ ] to be used to cite references in the text. Response: In the Vancouver Style, which is the required reference style for PLOS ONE, round brackets are used to cite references in the text. 

22. Funding support to be written in the acknowledgement. Response: PLOS guidelines indicate not to include funding sources in the Acknowledgements section of the paper. Funding information should only be entered in the financial disclosure section of the submission system. 

Reviewer 2 Feedback

1. My concerns about the manuscript primary relate to CONOSRT checklist items related to sample size (item 7a) and outcomes and estimation (items 17 a/b). In this pragmatic RCT, there were clearly described challenges in obtaining the required sample size to effectively measure the primary outcome. A sample of about 60% of the optimum number was obtained. The lack of significant findings may be attributed to the sample size. The authors included a discussion about the importance of reporting minimally important differences between groups, but it is not clear what the value of the minimally important differences are. Response: As indicated in the Outcomes and Measures section of the Methods, guidelines are available for judging clinical significance for the VR-12, but not the other outcome measures. VR-12 developers suggest a minimally important difference (MID) of 3 for interpreting group mean summary score differences (PCS, MCS). This is on Page 13 Line 307-309. 

2. While the authors present a clear and thoughtful report of a pragmatic RCT, the results appear to be quite inconclusive. The authors could consider examining some of the lessons learned from this research (e.g. measurement of depression, innovative recruitment methods) could be included a brief report versus a report of the RCT as a whole. Response: Because the results of this pragmatic RCT were inconclusive, in the Discussion section of the paper, we reflected on the lessons learned from this research, including the need to identify, test and implement innovative methods to enhance recruitment and retention of this population, and the limitations and challenges of implementing pragmatic trials. This in on Page 26-31. 

Reviewer 3 Feedback

Abstract 

1. Abstract, results: The authors state that having depressive symptoms along with multi-morbidity was an inclusion criterion. How is it possible then that only 56% of the participants had depressive symptoms? What does ‘clinically significant’ mean in this context? Please clarify. Response: We revised the Participants and Recruitment section of the Methods to indicate that older adults were eligible for the study if they screened positive for depressive symptoms as assessed by a two-item version of the Patient Health Questionnaire (PHQ-2). The purpose of the PHQ-2 is not to establish definitively the presence of a depressive disorder, but rather to screen for depressive symptoms as a “first step” approach (1). Thus, eligible, and consenting participants were further evaluated with the Center for Epidemiologic Studies Depression Scale 10-item tool (CESD-10) to determine whether they met the criteria for a depressive disorder (CES-D > 10). This change is on Page 8-9 Line 199-203. Depressive symptoms (> 10 on CES-D-10) were found in 72% of participants in both groups. This correction was made to the Baseline characteristics sub-section of the Results (Page 19 Line 452), and the proportion of participants with depressive symptoms was removed from the abstract, results. 

2. Abstract, results: As per the CONSORT guideline for abstracts of RCTs, the number of participants analyzed in each group needs to be stated. The authors only state how many participants they enrolled but not how many they ended up including in their analyses (by study group). Please add this information. This also raises the question whether the analyses were done per protocol, as treated, intention to treat? Please clarify. There are no details regarding the statistical analyses in the abstract at all. This would probably help clarify my next comment too. Response: The number of participants analyzed in each group was added to the abstract, results. We added a sentence to the abstract, outcome measures that intention-to-treat analysis was done using ANCOVA modeling. 

3. Abstract, results: Please clarify the following statement: “No significant group differences were seen for the baseline to six-month change in mental functioning (mean difference: 1.09, 95% CI: -3.24, 5.41, p=0.61)”. This sentence seems to mix a couple of things. If the authors compared group differences in baseline to follow-up CHANGES, then a simple mean difference is not the appropriate outcome. The way the authors present their results makes me think that they compared the follow up group means between the intervention group and the control group. A group comparison of baseline to follow up changes would require the comparison of the intervention group change (baseline to follow up mean difference in this group) to the control group change (baseline to follow up mean difference in that group). Please be clear about what was done. As per my comment above: a sentence or two related to the analytic approach in the methods section might clarify these issues and help the authors to present their results more clearly. The analyses are nicely described in the methods section of the paper. Response: We clarified in the Abstract, Outcome Measures and Results, and the Statistical Analysis Sub-Section of the Methods (Page 14, Line 330)that we compared the intervention group change (baseline to 6-months mean difference in this group) to the control group change (baseline to 6-months mean difference in this group). 

4. Abstract, conclusions: I think the conclusions are too strong. First, according to the results section in the abstract, only one outcome improved – but in the conclusions the authors speak about measures (plural). Second, receiving more information is not necessarily a patient experience measure. Whether patients receive more or less information does not indicate at all whether the information was appropriate and whether lack of information actually was the issue. Most importantly, it does not tell us anything about the patient’s experience. Therefore, I suggest the authors limit their statement to ‘patient information measures’. Stating that the intervention was cost neutral because detected differences were not statistically significant is also way too strong a statement, given that the study was under-powered to even detect differences in the main outcome – due to recruitment challenges. It was very likely not powered either to detect differences in this secondary outcome (costs). Therefore, the reason for statistically non-significant differences in costs may not be due to similar costs of the two interventions, but due to a lack of power (small sample size). Table 4 at least suggests that tendentially the total costs of the intervention exceeded those of usual care. Response: In the Abstract, conclusions, we clarified that the intervention resulted in improvements in one aspect of patient experience (information about health and social services). We revised the Discussion section of the paper to clarify that information provision is a component of patient experience. We indicated that: Information provision regarding available health and social services is a key aspect of patient’s care experience in the Intermediate Care for Older People Home-Based-Integrated Care Patient-Reported Experience Measure (IC-PREMs), that was used to measure patient experience in this study. In a recent systematic review of the reliability and validity of eighty-eight patient-reported experience measures, Bull et al. (2) reported several other tools that included information provision as a component of measuring patient experience. We agree, that whether patients receive more or less information does not indicate at all whether the information was appropriate and whether lack of information actually was the issue. This important point was also added to the discussion. These changes are on Page 30 Lines 652-657. We agree that the finding that there was no statistically significant difference between groups in per-person health service use costs may not be due to similar cost of the two interventions, but due to a lack of power (small sample size). We removed the sentences in the abstract, conclusions, and the discussion (Page 26 Line 569-570) that stated that the intervention was cost neutral and added a sentence in the discussion to indicate that the finding that there was no statistically significant difference between groups in per-person health service use costs may not be due to similar cost of the two interventions, but due to a lack of power (small sample size) (Page 28 Line 614-616).

5. Abstract: no study registration number available in the abstract (although required according to the CONSORT guideline for abstracts of RCTs). Response: The clinicaltrials.gov registration number was added to the abstract in accordance with the CONSORT guidelines. 

Introduction 

6. In lines 108 and following, the authors make a case for why this specific intervention they tested is needed. However, I think some more specifics are required here. The authors state that multiple studies are available demonstrating poor quality of ‘care transitions. I think what the authors refer to are transitions from an acute care hospital to the community – not any of the other possible healthcare transitions (e.g., hospital-rehab, hospital-LTC, community-hospital, etc.). Please clarify and be consistent. Response: We clarified in the Introduction section of the paper that we are referring to “hospital-to-home” transitions. 

7. More importantly, the study reported here is set in Canada but only one of the three studies cited is a Canadian study, the other two studies are Norwegian. Please elaborate on the concrete issues present and to be addressed in the Canadian context (or more precisely in Ontario where the study was situated). The following arguments about factors contributing to fragmented and poor-quality transitions also largely cite US studies, a few are European, and many are more than 10 years old. I am not sure whether these studies are most suitable to inform an intervention study that addresses issues in a Canadian context – at least not without adding evidence that the exact issues and contributing factors that international studies found are present in the Canadian context too. Otherwise, an intervention study addressing these factors may not address the right causes and therefore may not be effective. Response: This is an excellent point. Response: We removed references to studies that were more than 10 years old and added several studies conducted in a Canadian setting (including Ontario where the study was situated) to support the identification of factors in Canada, the USA, and elsewhere that contribute to poor quality hospital-to-home transitions and adverse outcomes. This change is on Page 5 Line 109-126. 

8. Also, in their study protocol, the authors state that they used components that were demonstrated to be effective in previous transitional interventions (e.g., home visits, telephone contacts, care coordination) and they cite studies demonstrating that these previous interventions were effective. If there are effective transitional interventions, why not use an existing one and adapt it to the specific context conditions Ontario, Canada? I think the authors do not sufficiently justify why a new intervention is needed, rather than building on an existing one. Response: We provided more detail in the Introduction section of the paper regarding the gaps that exist in the literature regarding the effectiveness of transitional care interventions for older adults with multimorbidity and depressive symptoms. We clarified in the paper that although key elements of effective hospital-to-home transitional care interventions have been identified, what elements should be included in transitional care interventions for older adults with stroke and multimorbidity and depressive symptoms remains inconclusive. This change is on Page 5-6 Line 130-148. 

Methods 

9. Methods, intervention description: Neither in their trial protocol nor in this results paper do authors give any details on how the intervention was developed – i.e. based on which theoretical foundations, evidence, and methods to develop complex interventions (e.g. the MRC framework, the Behaviour Change Wheel, Intervention Mapping, etc.). Also, from the intervention description it is unclear whether, to what extent and how people with lived experience (patients and their families) and decision makers were involved in developing the intervention. I suggest moving the patient & public involvement section up, right after (or before) the intervention description – if allowed by the author instructions. Response: We added more detail in the Introduction section and the Intervention sub-section of the Methods on how the intervention was developed, and described how a range of stakeholders, including patients and their families with lived experience, providers and decision-makers were involved in developing the intervention. This change is on Page 9 Line 213-224. We also moved the patient and public involvement section up, right after the intervention description. This change is on Page 11 Line 268-280. 

10. Methods, outcome measures: in their study protocol, as well as, in this results paper, the authors merely list the tools used to assess study outcomes – and while references are cited that may include results on the tools’ psychometric properties, the authors do not mention at all for any of the tools used what is known about their reliability and validity. This could be done in an appendix if the authors must save space, but robust psychometric properties of research tools are a key component of rigour in a trial like this one – and they have to be reported. Response: We added a sentence to the Outcomes and Measures sub-section of the Methods to indicate that all the tools used to assess the study outcomes were reliable and valid and have been used in our previous trials involving community-living older adults with multimorbidity (Page 12-13 Line 297-299). We checked and updated all the references to these tools that support the tools’ reliability and validity (Page 12-13 Line 286-303). 

Results 

11. Results, baseline characteristics: same comment as in abstract: the authors need to define what ‘clinically significant’ depressive symptoms mean, compared to clinically non-significant symptoms. Also please explain why not everyone had depressive symptoms if having these symptoms was an inclusion criterion. Most importantly, the authors report that 56% of participants had clinically significant depressive symptoms in the abstract while here they report that 72% of participants had clinically significant depressive symptoms. Even more confusing, further down the proportion of participants with clinically significant depressive symptoms is 69%. If the discrepancies between the abstract and the baseline characteristics section are because the abstract reports numbers at follow up, please be clear about that – and definitely clarify the two different proportions of participants with clinically significant depressive symptoms in this section (I think 72% is the number for both study groups taken together, 69% is the proportion in the intervention group). Response: We apologize for the inconsistencies in the text in reporting the prevalence of depressive symptoms. As previously mentioned, we revised the Participants and Recruitment section of the Methods to indicate that older adults were eligible for the study if they screened positive for depressive symptoms as assessed by a two-item version of the Patient Health Questionnaire (PHQ-2). The purpose of the PHQ-2 is not to establish definitively the presence of a depressive disorder, but rather to screen for depressive symptoms as a “first step” approach (1). Thus, eligible, and consenting participants were further evaluated with the Center for Epidemiologic Studies Depression Scale 10-item tool (CESD-10) to determine whether they met the criteria for a depressive disorder (CES-D > 10) (Page 8=9 Line 199-203). Depressive symptoms (> 10 on CES-D-10) were found in 72% of participants at baseline in both groups. This correction was made to the Baseline characteristics sub-section of the Results, and the reference to 69% of participants with depressive symptoms was removed along with the proportion of participants who had depressive symptoms or were taking an antidepressant medication This change is on Page 19-20 Line 454-459. 

Discussion 

12. Discussion, first paragraph: same comments as in abstract – the conclusions are too strong. Neither is it justified to state that patient experience improved (since more information is not a proxy for good patient experience), nor does an under-powered study allow to conclude that the lack of statistical significance suggests that costs between the intervention and usual care did not differ. The same is true for lines 576 and following (more information does not equal better quality of care or better patient experience – more is not necessarily better). Response: As mentioned previously in our response to this reviewers’ comments in the abstract section, we revised the first paragraph of the discussion section to indicate that only one aspect of patient experience improved (Page 26 Line 568). We revised the Discussion section of the paper to clarify that information provision is a component of patient experience: Information provision regarding available health and social services is a key aspect of patient’s care experience in the Intermediate Care for Older People Home-Based-Integrated Care Patient-Reported Experience Measure (IC-PREMs), that was used to measure patient experience in this study. In a recent systematic review of the reliability and validity of eighty-eight patient-reported experience measures, Bull et al.(2) reported several other tools that included information provision as a component of measuring patient experience (Page 30 Line 652-656). We agree that whether patients receive more or less information does not indicate at all whether the information was appropriate and whether lack of information actually was the issue. We added this important point to the Discussion section of the paper (Page 30 Line 656-658). We added a sentence to the discussion to indicate that the finding that there was no statistically significant difference between groups in per-person health service use costs may not be due to similar cost of the two interventions, but due to a lack of power (small sample size) (Page 28 Line 614-616). We removed the sentence in the discussion that stated that the intervention was cost neutral (Page 26 Line 569-570)).

13. Discussion, lines 501, 502: now the authors state again that the baseline rate of clinically significant depressive symptoms was 56%, which is different from the numbers reported in the results section. In line 510 the number is back to 69%. I think the 69% refers to people with either clinically significant depressive symptoms or anti-depressant use, but please clearly explain and align those numbers throughout the paper. Response: We corrected the sentence in the Discussion section to indicate that the baseline rate of depressive symptoms in the present sample was 72% so that it is consistent with the rate reported in the Results section (Page 27 Line 578) We also removed the paragraph in the Discussion section reporting that 69% had depressive symptoms or were taking an antidepressant (Page 27 Line 586-593). 

14. Discussion, limitations – recruitment/retention challenges: I highly appreciate the efforts and resources the authors invested to recruiting participants and I can only imagine the struggles and frustration encountered. It is highly valuable to publish studies that openly report these challenges so other research teams can be better prepared for the issues they will encounter. It is also highly valuable that the authors are currently preparing a publication on the process evaluation findings of their trial. Systematically assessing what worked and what did not – and possible mitigation strategies – will be an important contribution to the literature. While I understand that the authors want to do more work on these issues and publish those results separately, I would still appreciate if they could add a couple of key learnings to the discussion section – i.e. what would they do differently now and what advice do they have for other research teams who intend to do intervention research with this population. Response: Because the results of this pragmatic RCT were inconclusive, in the Discussion section of the paper, we reflected on the lessons learned from this research, including the need to identify, test and implement innovative methods to enhance recruitment and retention of this population, and the limitations and challenges of implementing pragmatic intervention studies with this population. As the reviewer pointed out, we will be submitting a paper for publication on the results of the implementation evaluation that will provide further details about these lessons learned. 

Conclusions

15. Conclusions: same comments as to the abstract and the discussion sections – wording needs to be revised so it adequately reflects what can be concluded from this study. Response: The wording was revised so that it reflects what can be concluded from this study. Specifically, we removed the sentence in the discussion that stated that the intervention was cost neutral relative to usual care (Page 26 Line 569-570). We also clarified that the intervention resulted in improvements in only one aspect of patient experience (Page 26 Line 568).

1. Kroenke K, Spitzer RL, Williams JB. The Patient Health Questionnaire-2: validity of a two-item depression screener. Medical care. 2003;41(11):1284-92.

2. Bull C, Byrnes J, Hettiarachchi R, Downes M. A systematic review of the validity and reliability of patient-reported experience measures. Health Services Research. 2019;54(5):1023-35.

---

## [Decision Letter · Decision Letter 1]

30 Jun 2021

Effectiveness of a nurse-led hospital-to-home transitional care intervention for older adults with multimorbidity and depressive symptoms: A pragmatic randomized controlled trial

PONE-D-20-18099R1

Dear Dr. Markle-Reid,

We’re pleased to inform you that your manuscript has been judged scientifically suitable for publication and will be formally accepted for publication once it meets all outstanding technical requirements.

Kind regards,

Samuel Yeung-shan Wong

Academic Editor

PLOS ONE

Additional Editor Comments (optional):

Reviewers' comments:

Reviewer's Responses to Questions

**Comments to the Author**

1. If the authors have adequately addressed your comments raised in a previous round of review and you feel that this manuscript is now acceptable for publication, you may indicate that here to bypass the “Comments to the Author” section, enter your conflict of interest statement in the “Confidential to Editor” section, and submit your "Accept" recommendation.

Reviewer #1: (No Response)

Reviewer #2: (No Response)

Reviewer #3: All comments have been addressed

2. Is the manuscript technically sound, and do the data support the conclusions?

Reviewer #1: Partly

Reviewer #2: Yes

Reviewer #3: Yes

3. Has the statistical analysis been performed appropriately and rigorously? 

Reviewer #1: (No Response)

Reviewer #2: I Don't Know

Reviewer #3: Yes

4. Have the authors made all data underlying the findings in their manuscript fully available?

Reviewer #1: Yes

Reviewer #2: No

Reviewer #3: No

5. Is the manuscript presented in an intelligible fashion and written in standard English?

Reviewer #1: Yes

Reviewer #2: Yes

Reviewer #3: Yes

6. Review Comments to the Author

Reviewer #1: (No Response)

Reviewer #2: In the introduction, especially in the first paragraph, the text could be simplified to more clearly state the importance and rationale for this study, especially considering that the text was modified significantly from the initial submission.

In the discussion, it would be helpful to recommend that qualitative research on this topic is warranted and this could be a recommendation made for future research, especially related to patient experience and how this could contribute to improving health care for older adults.

Reviewer #3: Thank you for the authors for thoroughly and conclusively addressing all my comments. One last minor thing I noticed is that in the discussion section it is mentioned twice that the intervention was cost neutral. While the authors removed one of these statements, the second statement (p. 30, line 664 is still included). This can be addressed in the editing process and - apart from that last point - I recommend to accept the manuscript in its current form.

7. PLOS authors have the option to publish the peer review history of their article (what does this mean?). If published, this will include your full peer review and any attached files.

Reviewer #1: No

Reviewer #2: No

Reviewer #3: **Yes: **Matthias Hoben

---

## [Editor Report · Acceptance letter]

16 Jul 2021

PONE-D-20-18099R1 

Effectiveness of a nurse-led hospital-to-home transitional care intervention for older adults with multimorbidity and depressive symptoms: A pragmatic randomized controlled trial 

Dear Dr. Markle-Reid:

I'm pleased to inform you that your manuscript has been deemed suitable for publication in PLOS ONE. Congratulations! Your manuscript is now with our production department. 

Kind regards, 

on behalf of

Dr. Samuel Yeung-shan Wong 

Academic Editor

PLOS ONE